# LogicAttack: Adversarial Attacks for Evaluating Logical Consistency of Natural Language Inference

**Mutsumi Nakamura**[*]     **Santosh Mashetty**[*]     **Mihir Parmar**[*]
**Neeraj Varshney**     **Chitta Baral**

School of Computing and AI, Arizona State University
{mutsumi, smashett, mparmar3, nvarshn2, chitta}@asu.edu

## Abstract

Recently Large Language Models (LLMs) such as GPT-3, ChatGPT, and FLAN have led to impressive progress in Natural Language Inference (NLI) tasks. However, these models may rely on simple heuristics or artifacts in the evaluation data to achieve their high performance, which suggests that they still suffer from logical inconsistency. To assess the logical consistency of these models, we propose a LogicAttack, a method to attack NLI models using diverse logical forms of premise and hypothesis, providing a more robust evaluation of their performance. Our approach leverages a range of inference rules from propositional logic, such as Modus Tollens and Bidirectional Dilemma, to generate effective adversarial attacks and identify common vulnerabilities across multiple NLI models. We achieve an average $\sim 53\%$ Attack Success Rate (ASR) across multiple logic-based attacks. Moreover, we demonstrate that incorporating generated attack samples into training enhances the logical reasoning ability of the target model and decreases its vulnerability to logic-based attacks [1].

## 1 Introduction

Recently, LLMs have demonstrated impressive performance on Natural Language Inference (NLI) tasks (Raffel et al., 2020). Prior to the introduction of prompt paradigm, the top-performing NLI systems relied heavily on pre-training, followed by fine-tuning on labeled task-specific data (Nie et al., 2020a). However, with the emergence of prompt-based LLMs, these models have achieved good performance ($\sim 90\%$) on NLI tasks with minimal training data (i.e., few-shot learning) (Liu et al., 2023). These systems have demonstrated robustness to variations on syntax and lexical level, and the ability to generalize beyond training data

(Brown et al., 2020). However, a crucial question remains: do these models generate a logical output on NLI, or do they rely on simple heuristics and learn shortcuts to generate their results? Hence, we believe that logical reasoning is a critical component of NLI systems and should be subject to evaluation.

Here, we aim to address an important question: "Are widely used NLI models exhibit logical consistency?" Although attempts have been made to create adversarial attacks for NLI (Williams et al., 2022; Chien and Kalita, 2020), and also to evaluate the logical consistency of LLMs across a range of tasks (Gaskell et al., 2022), they do not consider logic-based attack generation[2]. To bridge this gap, we propose LogicAttack, a method to create logic-based adversarial attacks that utilize a variety of inference rules from propositional logic (PL). Here, we utilize PL since pre-training data (usually data from the web or books) lacks sufficient explicit relations (e.g., implies, and, not, or, etc.) between propositions in different inference rules, making it essential to evaluate reasoning ability of models on such logic-based relationships. Specifically, we employ six inference rules to generate attacks from PL. Table 1 depicts example instances from the SNLI (Bowman et al., 2015) on which the LogicAttack technique is applied to create an adversarial attack. In Table 1, we give examples of some perturbed (i.e., adversarial) samples created through the application of logical rules. These samples were generated using *(premise, hypothesis)* pairs with "entailment" labels exclusively.

In this work, we evaluate range models, including both fine-tuned models on NLI such as RoBERTa (Liu et al., 2019a), and BART (Lewis et al., 2020); and prompt-based models such as GPT-4, GPT-3 (Brown et al., 2020), ChatGPT and FLAN-T5 (Chung et al., 2022), using LogicAttack. Experimental results show that the logic-

---

[1]Data and source code are available at `https://github.com/msantoshmadhav/LogicAttack`

[*]Equal Contribution

[2]Detailed related work is discussed in Appendix A

| Name | Formal Expressions | Attack Examples |
|---|---|---|
| Original 1 | $(p_1 \rightarrow h_1)$ | **Premise:** An older man wearing a salon drape getting a haircut. 
 **Hypothesis:** A man gets a haircut. |
| Original 2 | $(p_2 \rightarrow h_2)$ | **Premise:** The dog runs towards the ball. 
 **Hypothesis:** A dog runs. |
| Modus Tollens | $((p_1 \rightarrow h_1) \wedge \neg h_1) \vdash \neg p_1$ | **Premise:** If an older man wearing a salon drape getting a haircut, then a man gets a haircut. No man gets a haircut. 
 **Hypothesis:** No older man wearing a salon drape is getting a haircut. |
| Constructive Dilemma | $((p_1 \rightarrow h_1) \wedge (p_2 \rightarrow h_2) \wedge (p_1 \vee p_2)) \vdash (h_1 \vee h_2)$ | **Premise:** If an older man wearing a salon drape getting a haircut, then a man gets a haircut. And if the dog runs towards the ball, then a dog runs. But either an older man wearing a salon drape getting a haircut or the dog runs towards the ball. 
 **Hypothesis:** Either a man gets a haircut or a dog runs. |
| Destructive Dilemma | $((p_1 \rightarrow h_1) \wedge (p_2 \rightarrow h_2) \wedge (\neg h_1 \vee \neg h_2)) \vdash (\neg p_1 \vee \neg p_2)$ | **Premise:** If an older man wearing a salon drape getting a haircut, then a man gets a haircut. And if the dog runs towards the ball, then a dog runs. But either no man gets a haircut or no dog runs. 
 **Hypothesis:** Either no older man wearing a salon drape is getting a haircut or the dog does not run towards the ball. |
| Bidirectional Dilemma | $((p_1 \rightarrow h_1) \wedge (p_2 \rightarrow h_2) \wedge (p_1 \vee \neg h_2)) \vdash (h_1 \vee \neg p_2)$ | **Premise:** If an older man wearing a salon drape getting a haircut, then a man gets a haircut. And if the dog runs towards the ball, then a dog runs. But either an older man wearing a salon drape getting a haircut or no dog runs. 
 **Hypothesis:** Either a man gets a haircut or the dog does not run towards the ball. |
| Transportation 1 | $((p_1 \rightarrow h_1) \vdash (\neg h_1 \rightarrow \neg p_1)$ | **Premise:** If an older man wearing a salon drape getting a haircut, then a man gets a haircut. 
 **Hypothesis:** If no man gets a haircut, then no older man wearing a salon drape is getting a haircut. |
| Transportation 2 | $(\neg h_1 \rightarrow \neg p_1) \vdash ((p_1 \rightarrow h_1)$ | **Premise:** If no man gets a haircut, then no older man wearing a salon drape is getting a haircut. 
 **Hypothesis:** If an older man wearing a salon drape getting a haircut, then a man gets a haircut. |
| Material Implication 1 | $(p_1 \rightarrow h_1) \vdash (\neg p_1 \vee h_1)$ | **Premise:** If an older man wearing a salon drape getting a haircut, then a man gets a haircut. 
 **Hypothesis:** Either no older man wearing a salon drape is getting a haircut or a man gets a haircut. |
| Material Implication 2 | $(\neg p_1 \vee h_1) \vdash (p_1 \rightarrow h_1)$ | **Premise:** Either no older man wearing a salon drape is getting a haircut or a man gets a haircut. 
 **Hypothesis:** If an older man wearing a salon drape getting a haircut, then a man gets a haircut. |
| Negate Hypothesis | $(p_1 \rightarrow \neg h_1)$ | **Premise:** An older man wearing a salon drape getting a haircut. 
 **Hypothesis:** No man gets a haircut. |

Table 1: Example of all nine adversarial attacks generated by utilizing six inference rules from propositional logic and negation. Here, Original 1 and Original 2 represent two data instances from the evaluation set of SNLI.

based attacks significantly reduce model performance, achieving higher ASR of $\sim 50\%$ on SNLI and $\sim 55\%$ on MNLI, indicating that these models rely on simple heuristics rather than using desirable reasoning processes. Additionally, we enhance the robustness of RoBERTa by fine-tuning a limited set of attack samples ($\sim 9k$) created from SNLI. Results demonstrate that fine-tuning the model on these samples maintains its accuracy on the original evaluation set and significantly decreases the ASR on SNLI. This indicates that LogicAttack is useful for the robust evaluation of NLI models and for enhancing their logical capabilities. Furthermore, our analysis of results leads to several interesting findings. Overall, this study highlights that models still rely on simple heuristics for generating outputs for NLI, emphasizing the need to enhance their logical reasoning capabilities.

## 2 LogicAttack

This section provides a detailed description of task formulation, our attack strategies, and algorithm.

### 2.1 Task Formulation

NLI models train to learn $f : (p, h) \rightarrow y$, where $p$ denoted premise, $h$ denotes hypothesis, and $y \in$ {entailment, contradiction, neutral}. Here, we probe NLI models using an adversarial attack by

perturbing $p$, and $h$. We leverage various inference rules from the PL to perturb $(p, h)$ pair having entailment relation into $(p', h')$ keeping the entailment relationship, but being able to fool many models to infer otherwise. The process of creating $(p', h')$ from $(p, h)$ using different rules is denoted as LogicAttack. By generating a set of diverse perturbations ($\mathcal{P}$ where $\forall (p', h') \in \mathcal{P}$), we aim to maximize the likelihood of fooling the model.

### 2.2 Attack Strategies

In order to generate a diverse set of perturbations, we carefully select six inference rules from PL: (1) Modus Tollens, (2) Constructive Dilemma, (3) Destructive Dilemma, (4) Bidirectional Dilemma, (5) Transposition, and (6) Material Implication. Additionally, we investigate the potential for generating adversarial attacks by negating the hypothesis ($h$) (denoted as "Negate Hypothesis"). A formal representation of each inference rule and corresponding examples of generated adversarial attacks is presented in Table 1. Moreover, a detailed explanation of each inference rule and negate hypothesis method is presented in Appendix B.

### 2.3 Attack Algorithm

Here, we present Algorithm 1 that demonstrates the procedural steps to perform LogicAttack using Modus Tollens on any NLI model ($\mathcal{M}$). The aim is

to have $(p, h)$ pair such that $p$ entails $h$, then apply Modus Tollens to generate perturbed $(p', h')$ pair. Note that, perturbations using inference rules such as Modus Tollens will not alter its entailment label.

---

**Algorithm 1** LogicAttack using Modus Tollens
---
1: **Input:** NLI Model $\mathcal{M} : f(p, h) \to y$, Evaluation set $\mathcal{T}$
2: **Output:** $\mathcal{A}$ (Attack Success Rate)
3: **Function:** $Negation(s)$ (returns negation of sentence $s$)
4: **for** $(p, h, y) \in \mathcal{T}$ **do**
5:     **if** $y ==$ Entailment **then**
6:         **if** $f(p, h) ==$ Entailment **then**
7:             $premise(p, h) = ((p \to h) \wedge Negation(h))$
8:             $hypothesis(p, h) = Negation(p)$
9:             $p' \leftarrow premise(p, h)$
10:            $h' \leftarrow hypothesis(p, h)$
11:            $\mathcal{M} : f(p', h') \to \hat{y}$
12:            **if** $\hat{y} \mathrel{!=}$ Entailment **then**
13:                $InCorrect \leftarrow InCorrect + 1$
14: $\mathcal{A} \leftarrow \frac{InCorrect}{length(\mathcal{T})}$

---

In Algorithm 1, $(p, h, y)$ denotes the original pair of premise, hypothesis, and label; $(p', h')$ denotes perturbations; and $\hat{y}$ denotes model generated label for $(p', h')$. Additionally, Algorithm 1 incorporates the function $Negation(s)$, which is responsible for generating the negation of any given sentence $(s)$. In this work, we prompt GPT-3 to generate negations. Note that, generating the negation of logical sentences can be complicated, hence, it is important to provide multiple in-context examples of how negated sentences should be generated for different logical forms. To get a better sense of the correctness of our method, 500 random sentences from SNLI were negated using $Negation(s)$ and reviewed by an author. The precision for these negations is $\sim 98\%$. The prompt used for generating negation is presented in Appendix C.

In Algorithm 1, $premise(p, h)$ and $hypothesis(p, h)$ play a crucial role in creating logic-based attacks. To illustrate this, let's consider an example where $p$ is "An older man wearing a salon drape getting a haircut.", $h$ is "A man gets a haircut.", and $y$ is "Entailment", thus $(p \to h)$. Now, we apply the Modus Tollens (formally expressed as $((p \to h) \wedge \neg h) \vdash \neg p$), and get a $(p', h')$. The resulting $p'$ is "If an older man wearing a salon drape getting a haircut, then a man

gets a haircut. No man gets a haircut.", and $h'$ is "No older man wearing a salon drape is getting a haircut." Algorithm 1 provides a demonstration of LogicAttack using the Modus Tollens. However, a similar algorithm applies to generate attacks for other inference rules illustrated in Table 1.

# 3 Experiments and Results

## 3.1 Experimental Setup

**Dataset** To generate attacks (Algorithm 1), we utilize the evaluation set of SNLI (Bowman et al., 2015) and MNLI (Williams et al., 2018a). In particular, we only generate attacks for the premise and hypothesis pairs with the "Entailment" label. In addition, we use $1k$ entailment instances from the SNLI train set to generate attack samples for training purposes.

**Models** We evaluate models on three different configurations: (i) single-task, (ii) multi-task, and (iii) prompting. For single-task, we evaluate the RoBERTa-large fine-tuned on the SNLI and MNLI, respectively. For multi-task, we evaluate two models: RoBERTa-large, and BART-large fine-tuned on SNLI, MNLI, FEVER-NLI (Thorne et al., 2018), and ANLI (Williams et al., 2022). For prompt-based models, we evaluate GPT-4, GPT-3, Chat-GPT, and FLAN-T5 using zero-shot and few-shot prompts. Appendix C provides prompts and few-shot results are presented in Appendix D.

**Experiments** Here, we conduct two experiments: (i) evaluation of models using LogicAttack, and (ii) fine-tuning with attack instances. In (i), we evaluate each model on the evaluation data of SNLI and MNLI, and identify the samples where the model generates the correct label. Subsequently, we apply Algorithm 1 to these selected samples to create attacks, which are then used to evaluate the model's performance. In (ii), we aim to investigate the impact of incorporating logic-based attack sentences into the training set. To explore this, we generate $9k$ attack instances by randomly selecting $1k$ pairs from the training set of SNLI. We perform fine-tuning on the RoBERTa (large) model using both the attack samples and the original SNLI training set. More details about experiments are presented in Appendix E.

**Metric** As defined in (Gaskell et al., 2022), we are evaluating the performance of LogicAttack on each model using Attack Success Rate (ASR) and

| Dataset | Attack Type | Single-Task | Multi-Task | | Prompting | | | | F1 value |
|---------|-------------|-------------|------------|------|-----------|------|---------|-------|----------|
| | | RoBERTa | RoBERTa | BART | FLAN-T5 | GPT-3 | ChatGPT | GPT-4 | |
| **SNLI** | Modus Tollens | 99.7 | 94.1 | 45.5 | 99.9 | 97.8 | 96.3 | 50.6 | 61.0 |
| | Constructive Dilemma | 34.7 | 3.9 | 0.1 | 0.6 | 34.7 | 11.3 | 9.0 | 60.4 |
| | Destructive Dilemma | 93.6 | 30.2 | 24.9 | 87.3 | 90.4 | 46.7 | 18.8 | 57.9 |
| | Bidirectional Dilemma | 73.4 | 67.0 | 22.5 | 15.6 | 63.5 | 16.3 | 11.7 | 59.1 |
| | Transposition 1 | 99.7 | 82.5 | 99.1 | 82.0 | 79.6 | 25.0 | 22.7 | 58.6 |
| | Transposition 2 | 95.4 | 40.0 | 94.4 | 94.2 | 46.2 | 44.0 | 16.2 | 58.6 |
| | Material Implication 1 | 98.6 | 85.8 | 94.8 | 94.5 | 54.6 | 25.5 | 11.1 | 59.8 |
| | Material Implication 2 | 85.6 | 93.5 | 46.1 | 33.0 | 35.4 | 12.9 | 29.8 | 59.8 |
| | Negate Hypothesis | 1.1 | 3.2 | 1.9 | 0.6 | 35.0 | 15.4 | 9.7 | 95.2 |
| | **Avg.** | **76.1** | **56.2** | **48.7** | **56.4** | **59.7** | **32.7** | **20.0** | **63.4** |
| **MNLI** | Modus Tollens | 96.6 | 81.1 | 68.1 | 94.8 | 94.6 | 96.1 | 72.3 | 62.3 |
| | Constructive Dilemma | 23.8 | 2.8 | 0.7 | 7.0 | 26.9 | 4.6 | 0.0 | 62.2 |
| | Destructive Dilemma | 90.5 | 58.4 | 64.0 | 86.9 | 96.5 | 94.8 | 63.2 | 60.4 |
| | Bidirectional Dilemma | 85.6 | 68.9 | 52.8 | 49.4 | 70.1 | 58.4 | 30.6 | 61.3 |
| | Transposition 1 | 96.5 | 87.1 | 91.5 | 78.1 | 67.1 | 57.3 | 47.1 | 60.5 |
| | Transposition 2 | 90.4 | 74.1 | 92.3 | 95.4 | 33.6 | 32.7 | 42.9 | 60.5 |
| | Material Implication 1 | 88.9 | 70.4 | 72.9 | 78.6 | 45.6 | 53.1 | 40.7 | 61.4 |
| | Material Implication 2 | 75.8 | 79.3 | 64.9 | 43.2 | 18.9 | 7.6 | 69.9 | 61.4 |
| | Negate Hypothesis | 8.8 | 11.0 | 10.0 | 5.9 | 11.6 | 9.1 | 5.6 | 94.2 |
| | **Avg.** | **73.5** | **59.9** | **58.3** | **59.9** | **51.7** | **46.0** | **41.4** | **64.9** |

Table 2: Evaluation of different models in terms of ASR (%) using SNLI and MNLI. Higher ASR is better.

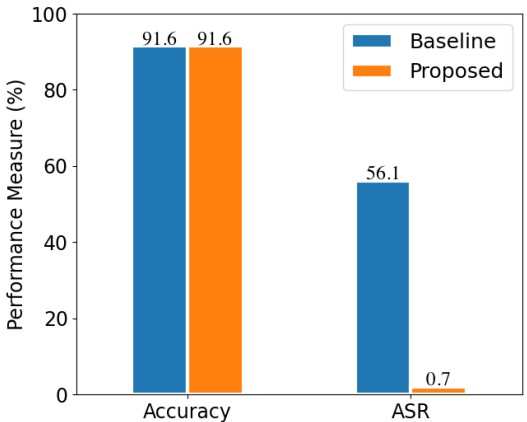

Figure 1: Performance of baseline (fine-tune with SNLI) and proposed (fine-tune with SNLI + attack samples) models. Higher accuracy and lower ASR is better.

F1 sentence overlap score (F1). To demonstrate the similarity between generated and original instances, we also incorporate SimCSE (Gao et al., 2021) and BERTScore (Zhang* et al., 2020) (results presented in the Appendix F).

### 3.2 Results and Analysis

Here, Table 2 represents ASR of each model evaluated using LogicAttack, and Figure 1 shows Accuracy and ASR of baseline and proposed models.

**ASR values tend to become higher when F1 values are lower.** From Table 2, we can observe that LogicAttack achieves a high ASR for the most of PL inference rules. Their F1 value averages

are between 58% and 62% and their difference is relatively small. However, the LogicAttack by "Negated Hypothesis" was unsuccessful on all models, and the average F1 value is $\sim 95\%$, meaning that sentences that have a high overlapping rate with the original sentences, yet just by negating the hypothesis, the models predict correctly on them. However, we do not seek to minimize F1 since low F1 is bad ("adversarial" implies minimal perturbations), and F1 value $\sim 60\%$ is preferable for effective attacks (Gaskell et al., 2022).

**# of negations *vs*. ASR** ASR tends to become higher when the ratio of (# of negated sentences / # of original sentences) is larger, i.e., the number of negated sentences in the premise/hypothesis is higher. Constructive, Destructive, and Bidirectional Dilemma utilize the same number of logical AND, logical OR, implications ($\rightarrow$), and original sentences, yet they have different ASR because of difference in the number of negations and where their negations appear, in the premise or hypothesis. Destructive Dilemma uses the maximum number of negations (four) in one pair of premise and hypothesis, among these logic rules, and also has the highest ASR. Modus Tollens also has a high ASR, however, Modus Tollens uses only four sentences from the SNLI test set while Destructive Dilemma uses eight sentences. Appendix G.1 presents statistics of negations in each inference rule.

**Larger model are less prone to attacks.** From Table 2, it is evident that as the size of the prompting model and the amount of pre-training data increase, ASR decreases. Moving from GPT-3 to GPT-4, there is a drop in ASR, where GPT-3 shows 59.7%, whereas GPT-4 shows 20.0% for SNLI, and 51.7% and 41.4% for MNLI, respectively. To investigate the improvements in mitigating ASR by GPT-4 compared to GPT-3, we randomly selected 20 instances in which GPT-4 succeeded while GPT-3 failed and applied the chain-of-thought prompting technique to analyze their reasoning steps for predicting the final answer. While evaluating the reasoning chain generated by GPT-4, we observed its ability to establish logical relations between the premise and hypothesis, thus performing logical inference to generate final answer. In contrast, GPT-3 simply compares premise and hypothesis (without any logical connections in majority of cases) for generating final answer, leading to a lack of logical consistency and, in some cases, contradictory results instead of entailment. Thus, GPT-4 has improved performance (low ASR) compared to GPT-3. However, GPT-4 still exhibits a relatively low ASR for certain inference rules (e.g., <50% for Modus Tollens). Further discussion (i.e., prompt and examples) is presented in Appendix G.2.

**Effect of Fine-tuning with Attack Samples** From Figure 1, it becomes evident that the model, when fine-tuned using a small set of attack samples ($\sim 9k$), achieves a lower ASR (0.7%) compared to the baseline (56.1%). Additionally, the proposed model maintains its performance on the original SNLI evaluation set (91.6%). This observation highlights that incorporating a small number of attack samples during training does not hinder the model's capacity to effectively perform the NLI task on the original (premise, hypothesis) pairs.

## 4 Conclusions

We introduce LogicAttack, a novel method for generating logic-based attacks to assess the robustness of NLI models. We evaluate a range of models on NLI, considering single-task, multi-task, and prompting. Experimental results demonstrate that these models are vulnerable to logic-based adversarial attacks, as evidenced by their higher ASR. Additionally, we investigate the impact of fine-tuning models with attack samples and observe a significant improvement in their logical reasoning capabilities, leading to a substantial decrease in ASR. Overall, our findings suggest that LogicAttack provides a valuable framework for conducting logically consistent evaluations of models.

## Limitations

In this work, we explore six inference rules from propositional logic, however, this study can be extended further to incorporate more inference rules to create logic-based attacks. Furthermore, we plan to extend our work to evaluate more models and also a range of other NLI datasets. In addition, this work only provides the logical consistent evaluation of NLI, however, this method can be extended to other natural language understanding tasks such as question-answering, and other reasoning tasks. Furthermore, the scope of this work is limited in terms of logical conjunctions and disjunctions where we use simple word "and" for conjunctions and "or" for disjunctions in our experiments. To this end, different ways to create logical conjunctions and disjunctions can be explored since it can be more complicated in the case of logic.

## Acknowledgement

We thank the anonymous reviewers for constructive suggestions, and the Research Computing (RC) at Arizona State University (ASU) for providing computing resources for experiments. We acknowledge support by NSF grant 2132724 and a 2023 Spring Amazon Research Award (ARA).

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

## A  Related Work

Natural Language Inference (NLI) is extensively studied in NLP and many datasets and models have been proposed to effectively perform NLI tasks (Camburu et al., 2018; Soares et al., 2023; Talman et al., 2021; Liu et al., 2022; Bowman et al., 2015; Williams et al., 2018b; Poliak, 2020). Varshney et al. (2022a) proposed a number of sentence transformations to procedurally create NLI training instances. To evaluate the robustness of these NLI models, one thread of work evaluates these models on out-of-domain datasets (Varshney et al., 2022b; Yang et al., 2023). Another thread aims at creating different adversarial attacks on NLI models (Williams et al., 2022; Chien and Kalita, 2020; Chan et al., 2020; Thorne et al., 2018). For instance, Adversarial NLI (Nie et al., 2020b) is one of the first attempts that introduces a dataset of carefully crafted adversarial examples to evaluate and assess the robustness and generalization capabilities of language models in the context of NLI tasks. Chien and Kalita (2020) highlights creating adversarial datasets to challenge NLI models which revealed failure in generalizing on unseen examples. Moreover, Chan et al. (2020) introduced *poison attack* technique to find vulnerability in NLI system.

However, these adversarial attack techniques do not consider logic-based techniques to create attack since logical reasoning is critical component of NLI system and subject to evaluation. Previous efforts have been made to assess the logical consistency of LLMs on range of other tasks. Most recent study by Gaskell et al. (2022) introduced a framework called LAVA which aims to address a problem of logical inconsistency by combining an adversarial and generative process for Soft Theorem-Proving (STP). However, Gaskell et al. (2022) do not utilize any logic-based inference rules to create adversarial attacks. Motivated by this, we propose LogicAttack, a framework to evaluate logical consistency of NLI models by creating logic-based attacks.

## B  Attack Strategies

Detailed explanation of inference rules employed to generate attacks[3]. Table 1 (main paper) provides examples of generated attack corresponding to each rule. To provide a more in-depth understanding of

---

[3]Part of the information in this section is adapted from https://en.wikipedia.org/wiki/Propositional_calculus

each attack, we introduce some notations: $p \to h$, where $p$ represents the premise, $h$ represents the hypothesis, and $\to$ denotes the "Entailment" relation between $p$ and $h$. Here, to generate attack using modus tollens, transposition, material implication, and negate hypothesis, we use only one $(p, h)$ pair to generate attacks. However, we use two $(p_1, h_1)$ and $(p_2, h_2)$ pairs to generate attack for constructive, destructive, and bidirectional Dilemma.

**Modus Tollens**  Modus Tollens is described as: "If $p$ then $h$; not $h$; therefore not $p$". Using this rule, we generate new premise, $p'$: $p \to h \land \neg h$, and new hypothesis, $h'$: $\neg p$.

**Transposition**  Transportation is described as: "If $p$ then $h$ is equivalent to if not $h$ then not $p$". We split this rule to two parts since it involves the equivalency: (i) Transportation1: using this rule, we generate new premise, $p'$: $p \to h$, and new hypothesis, $h'$: $\neg h \to \neg p$, and (ii) Transportation2: using this rule, we generate new premise, $p'$: $\neg h \to \neg p$, and new hypothesis, $h'$: $p \to h$.

**Material Implication**  Material Implication is described as: "If $p$ then $h$ is equivalent to not $p$ or $h$". Similar to transposition rule, we split this rule to two parts since it involves the equivalency: (i) Material Implication1: using this rule, we generate new premise, $p'$: $p \to h$, and new hypothesis, $h'$: $\neg p \lor h$, and (ii) Material Implication2: using this rule, we generate new premise, $p'$: $\neg p \lor h$, and new hypothesis, $h'$: $p \to h$.

**Negate Hypothesis**  Since we have pairs of two sentences $p$ and $h$, where "if $p$ then $h$", we negated its conclusion so that it will be "if $p$ then $\neg h$" and see if we can obtain the contradiction label. Here, we generate a new premise to be $p'$: $p$ and a new hypothesis to be $h'$: $\neg h$.

**Constructive Dilemma**  Constructive Dilemma is described as: "If $p_1$ then $h_1$; and if $p_2$ then $h_2$; but $p_1$ or $p_2$; therefore $h_1$ or $h_2$". Using this rule, we generate new premise, $p'$: $(p_1 \to h_1) \land (p_2 \to h_2) \land (p_1 \lor p_2)$, and new hypothesis, $h'$: $h_1 \lor h_2$.

**Destructive Dilemma**  Destructive Dilemma is described as: "If $p_1$ then $h_1$; and if $p_2$ then $h_2$; but not $h_1$ or not $h_2$; therefore not $p_1$ or not $p_2$". Using this rule, we generate new premise, $p'$: $(p_1 \to h_1) \land (p_2 \to h_2) \land (\neg h_1 \lor \neg h_2)$, and new hypothesis, $h'$: $\neg p_1 \lor \neg p_2$.

**Bidirectional Dilemma** Bidirectional Dilemma is described as: "If $p_1$ then $h_1$; and if $p_2$ then $h_2$; but $p_1$ or not $h_2$; therefore $h_1$ or not $p_2$". Using this rule, we generate new premise, $p'$: $(p_1 \rightarrow h_1) \wedge (p_2 \rightarrow h_2) \wedge (p_1 \vee \neg h_2)$, and new hypothesis, $h'$: $h_1 \vee \neg p_2$.

## C Prompts

### C.1 Negation Prompt

In order to create negated sentences, we have done multiple evaluations with GPT-3, Vicuna (Chiang et al., 2023) and Llama (Touvron et al., 2023) based models. According to the initial experiments, we found that GPT-3 was giving much better results. We handcrafted some negated sentence examples and included them as part of a prompt in GPT-3 to generate negated sentences. We sampled over 500 generated negated sentences and found their accuracy to be around around 98%. For negation generation, the parameters used for GPT-3: temperature = 0.7, max-tokens = 512, top-p = 1, frequency-penalty = 2, presence-penalty = 2.

The following is the prompt used to generate negated sentences:

> *sentence: A dog loves to play in the park.*
> *negation: No dog loves to play in the park.*
> *sentence: An apple is red.*
> *negation: No apple is red.*
> *sentence: Some dogs like to sleep on the grass.*
> *negation: No dog likes to sleep on the grass.*
> *sentence: Everybody is playing.*
> *negation: Somebody is not playing.*
> *sentence: Everyone likes sunny days.*
> *negation: Someone does not like sunny days.*
> *sentence: Every school has a math class.*
> *negation: Some schools do not have a math class.*
> *sentence: Every apple is sweet.*
> *negation: Some apples are not sweet.*
> *sentence: Jill is having a hot soup.*
> *negation: Jill is not having a hot soup.*
> *sentence: We don't watch the movie.*
> *negation: We watch the movie.*
> *sentence: He was there not long ago.*
> *negation: He was not there not long*

> *ago.*
> *sentence: It was so not good.*
> *negation: It was not so not good.*
> *sentence: She presented not very confidently.*
> *negation: She did not present not very confidently.*
> *sentence: It was a not great contribution.*
> *negation: It was a great contribution.*
> *sentence: They need a food that is bitter.*
> *negation: They don't need a food that is bitter.*
> *sentence: The children are playing.*
> *negation: The children are not playing.*
> *Provide a negation of the next sentence by following the examples above:*

### C.2 Prompts for Experiments

**Zero-Shot Prompt** The following is the format of a prompt used for the experiments using GPT-3, ChatGPT, and GPT-4 with zero-shot:

> *In this task, you are given two sentences (Premise and Hypothesis). Your task is to identify the Label (relation) between the given Premise and Hypothesis. If both sentences agree with each other then return "entailment"; if both sentences indicate opposite view from each other then return "contradiction"; and there is no relation between two sentences or relation can not be identified then return "neutral".*
>
> *Format:*
> *Premise: (Premise is a sentence that describes a condition on which a logical argument is based)*
> *Hypothesis: (Hypothesis is a sentence which is a plausible conjecture or explanation which can be proved or disproved)*
> *Label: entailment/contradiction/neutral*
>
> *Identify the Label (relation) between the following Premise and Hypothesis -*
> *Premise: this church choir sings to the masses as they sing joyous songs from the book at a church*
> *Hypothesis: the church is filled with song*

*Label:*

**3-Shots Prompt** The following is the format of a prompt used for the experiments using GPT-3, ChatGPT, and GPT-4 with 3-shots:

*In this task, you are given two sentences (Premise and Hypothesis). Your task is to identify the Label (relation) between the given Premise and Hypothesis. If both sentences agree with each other then return "entailment"; if both sentences indicate opposite view from each other then return "contradiction"; and there is no relation between two sentences or relation can not be identified then return "neutral"*

*Format:*
***Premise:*** *(Premise is a sentence that describes a condition on which a logical argument is based)*
***Hypothesis:*** *(Hypothesis is a sentence which is a plausible conjecture or explanation which can be proved or disproved)*
***Label:*** *entailment/contradiction/neutral*

***Premise:*** *A person on a horse jumps over a broken down airplane.*
***Hypothesis:*** *A person is training his horse for a competition.*
***Label:****neutral*
***Premise:*** *A person on a horse jumps over a broken down airplane.*
***Hypothesis:*** *A person is at a diner, ordering an omelette.*
***Label:****contradiction*
***Premise:*** *A person on a horse jumps over a broken down airplane.*
***Hypothesis:*** *A person is outdoors, on a horse.*
***Label:****entailment*

*By understanding the above examples, give the Label (relation) between the following sentences -*
***Premise:*** *this church choir sings to the masses as they sing joyous songs from the book at a church*
***Hypothesis:*** *the church is filled with song*
***Label:***

**6-Shots Prompt** The following is the format of a prompt used for the experiments using GPT-3, ChatGPT, and GPT-4 with 6-shots:

*In this task, you are given two sentences (Premise and Hypothesis). Your task is to identify the Label (relation) between the given Premise and Hypothesis. If both sentences agree with each other then return "entailment"; if both sentences indicate opposite view from each other then return "contradiction"; and there is no relation between two sentences or relation can not be identified then return "neutral"*

*Format:*
***Premise:*** *(Premise is a sentence that describes a condition on which a logical argument is based)*
***Hypothesis:*** *(Hypothesis is a sentence which is a plausible conjecture or explanation which can be proved or disproved)*
***Label:*** *entailment/contradiction/neutral*

***Premise:*** *A person on a horse jumps over a broken down airplane.*
***Hypothesis:*** *A person is training his horse for a competition.*
***Label:****neutral*
***Premise:*** *A person on a horse jumps over a broken down airplane.*
***Hypothesis:*** *A person is at a diner, ordering an omelette.*
***Label:****contradiction*
***Premise:*** *A person on a horse jumps over a broken down airplane.*
***Hypothesis:*** *A person is outdoors, on a horse.*
***Label:****entailment*
***Premise:*** *Children smiling and waving at camera*
***Hypothesis:*** *They are smiling at their parents*
***Label:****neutral*
***Premise:*** *Children smiling and waving at camera*
***Hypothesis:*** *There are children present*
***Label:****entailment*
***Premise:*** *Children smiling and waving at camera*

*Hypothesis: The kids are frowning*
*Label:contradiction*

*By understanding the above examples, give the Label (relation) between the following sentences -*
*Premise: this church choir sings to the masses as they sing joyous songs from the book at a church*
*Hypothesis: the church is filled with song*
*Label:*

## D  Few-shot Results

We have also experimented using GPT-3, ChatGPT, and GPT-4 with 3-shots and 6-shots, in addition to using zero-shot. Table 4 provides the results of few-shot experiments.

Giving examples of NLI as a few shots in a prompt of GPT-3, ChatGPT, and GPT-4 can enhance each model's ability to understand their tasks better, can improve their logical abilities, and resulted in lowering their ASRs (shown in Table 4). One thing to note is that changing from zero-shot to 3 shots improved each model's performance significantly, but changing from 3 shots to 6 shots did not. In some cases, it degraded their performance slightly. Another point is that the models seem to perform better for inference rules with longer premises and hypothesis (Constructive Dilemma, Destructive Dilemma, Bidirectional Dilemma).

## E  Experimental Setup

Here, we provide additional details about the experiments.

**Experiment (i)**    All models in this experiment are used from Huggingface except GPT-family models. For single-task, the model we used is RoBERTa-large fine-tuned on the SNLI[4]. For multi-task, the two models we used are: RoBERTa-large, and BART-large fine-tuned on SNLI, MNLI, FEVER, and ANLI[5]. For prompt-based models, the models used are: GPT-3 and GPT-4[6], ChatGPT [7], and

---

FLAN-T5 [8] using zero-shot and few-shot prompts. To perform adversarial attacks on GPT4, GPT3, and ChatGPT, the following were the parameters used: temperature = 0, max-tokens = 128, top-p = 1, frequency-penalty = 0, presence-penalty = 0. These parameters were used for all of GPT4, GPT3 and ChatGPT.

**Experiment (ii)**    During the fine-tuning process of RoBERTa (large)[9] (Liu et al., 2019b), the model is trained for 10 epochs, with a batch size of 16 and an initial learning rate of 5e-6. The experiments were conducted using NVIDIA GPUs, specifically the A6000 and A100 models. To establish a baseline for comparison, we also fine-tune the RoBERTa (large) model on the original training set with a similar configuration. Both models are then evaluated on the original evaluation set of SNLI, as well as the corresponding attack samples.

Here is a detailed explanation of the metrics used for evaluation:

- **Attack Success Rate (ASR)** This represents the attacker's performance: ASR = # successful attacks / # total attacks

- **F1 Sentence Overlap Score (F1)** F1 value was for each attack sample, an overlap score between the original (premise, hypothesis) pair and their corresponding perturbation generated by LogicAttack.

## F  SimCSE and BERTScore

We have also evaluated the resemblance between original and attack <premise, hypothesis> pairs using BERTScore[10] and SimCSE [11]. From Table 3, we can observe that BERTScore and SimCSE are 90% and 88% for all attack strategies (the score for each attack is average overall dataset samples). This shows the quality of generated perturbations.

## G  Qualitative Analysis

### G.1  Effect of Negations

Because all propositional logic rules used for logic attacks were generated from the same set of sentences in the SNLI test data set, we have checked their sentence structures to see what makes their

---

[4]https://huggingface.co/pepa/roberta-large-snli
[5]https://huggingface.co/ynie/roberta-large-snli_mnli_fever_anli_R1_R2_R3-nli
[6]https://platform.openai.com/
[7]https://chat.openai.com/

[8]https://huggingface.co/docs/transformers/model_doc/flan-t5
[9]https://huggingface.co/roberta-large
[10]https://github.com/Tiiiger/bert_score
[11]https://github.com/princeton-nlp/SimCSE

| Dataset | Attack Strategies | SimCSE | BERTScore |
|---------|-------------------|--------|-----------|
| SNLI | Modus Tollens | 0.839 | 0.919 |
| | Constructive Dilemma | 0.935 | 0.908 |
| | Destructive Dilemma | 0.869 | 0.906 |
| | Bidirectional Dilemma | 0.909 | 0.906 |
| | Transposition 1 | 0.866 | 0.912 |
| | Transposition 2 | 0.848 | 0.904 |
| | Material Implication 1 | 0.901 | 0.917 |
| | Material Implication 2 | 0.880 | 0.910 |
| | Negate Hypothesis | 0.868 | 0.967 |
| | **Avg.** | **0.879** | **0.917** |
| MNLI | Modus Tollens | 0.846 | 0.926 |
| | Constructive Dilemma | 0.937 | 0.914 |
| | Destructive Dilemma | 0.892 | 0.914 |
| | Bidirectional Dilemma | 0.914 | 0.914 |
| | Transposition 1 | 0.864 | 0.920 |
| | Transposition 2 | 0.864 | 0.911 |
| | Material Implication 1 | 0.905 | 0.925 |
| | Material Implication 2 | 0.891 | 0.918 |
| | Negate Hypothesis | 0.879 | 0.963 |
| | **Avg.** | **0.888** | **0.923** |

Table 3: SimCSE and BERTScore between original and attack <premise, hypothesis> pairs from SNLI and MNLI.

ASR different. Their main differences are the number and the location (premise or hypothesis) of certain logical words such as "and", "or", and "if .. then .." (implication), and also the number of included sentences from the SNLI test data. The number of negated sentences is also an important factor because they are modified from the original sentences, and can lower F1 values. The information in the Table 5 was used for the ASR analysis. In terms of their location, i.e., whether they appear in the premise or hypothesis, it also affects their ASRs. Between Transposition 1 and Transposition 2, or Material Implication 1 and Material Implication 2, only their premise and hypothesis are swapped, however, their ASRs differ. The main difference caused by swapping them is that in Transposition 2 and Material Implication 2, negated sentences appear in their premise instead of hypothesis.

## G.2 Prompt for GPT-4 *vs.* GPT-3

We conducted an analysis of the drop in ASR performance between GPT-4 and GPT-3, investigating the reasoning process employed by each model to provide answers. To achieve this, we utilized the following chain-of-thought prompt for our study:

*This is a Natural Language Inference task. Given the premise and hypothesis that contains rules of logical reasoning in natural language, perform step-by-step reasoning to predict one of three labels: Entailment, Contradiction, or Neutral. Please use the below format:*

*Premise: [natural language text for premise]*
*Hypothesis: [natural language text for hypothesis]*
*Reasoning steps: [generate step-by-step reasoning]*
*Answer: Entailment/Neutral/Contradiction*

*Premise:*
*Hypothesis:*
*Answer:*

| Inference Rules | GPT-3 | | ChatGPT | | GPT-4 | |
|---|---|---|---|---|---|---|
| | 3-shots | 6-shots | 3-shots | 6-shots | 3-shots | 6-shots |
| Modus Tollens | 69.4 | 67.2 | 90.9 | 92.8 | 12.3 | 11.4 |
| Constructive Dilemma | 2.4 | 4.2 | 3.7 | 3.6 | 5.3 | 5.8 |
| Destructive Dilemma | 15.9 | 15.6 | 14.7 | 17.7 | 13.8 | 13.6 |
| Bidirectional Dilemma | 4.3 | 6.5 | 5.4 | 6.1 | 8.3 | 8.6 |
| Transposition 1 | 71.5 | 68.7 | 4.7 | 5.3 | 13.2 | 10.0 |
| Transposition 2 | 11.5 | 16.1 | 10.3 | 11.7 | 12.8 | 11.1 |
| Material Implication 1 | 5.1 | 6.5 | 3.7 | 4.1 | 5.6 | 5.9 |
| Material Implication 2 | 4.1 | 7 | 3.7 | 3.7 | 23.5 | 15.9 |
| Negate Hypothesis | 4.4 | 5.6 | 7.9 | 8.7 | 5.7 | 6.4 |
| Avg. | 19.5 | 20.8 | 16.1 | 17.1 | 11.2 | 9.9 |

Table 4: Evaluation using GPT-3, ChatGPT, GPT-4 with 3 shots and 6 shots w.r.t. ASR (%)

| Inference Rules | # of negations | # of logical AND | # of logical OR | # of implications | # of contained original sentences |
|---|---|---|---|---|---|
| Modus Tollens | 2 | 1 | 0 | 1 | 4 |
| Constructive Dilemma | 0 | 2 | 2 | 2 | 8 |
| Destructive Dilemma | 4 | 2 | 2 | 2 | 8 |
| Bidirectional Dilemma | 2 | 2 | 2 | 2 | 8 |
| Transposition 1 | 2 | 0 | 0 | 2 | 4 |
| Transposition 2 | 2 | 0 | 0 | 2 | 4 |
| Material Implication 1 | 1 | 0 | 1 | 1 | 4 |
| Material Implication 2 | 1 | 0 | 1 | 1 | 4 |
| Negate Hypothesis | 1 | 0 | 0 | 0 | 2 |

Table 5: Statistics of generated sentences in terms of various characteristics w.r.t. inference rules.