# OpenReview forum: "LogicAttack: Adversarial Attacks for Evaluating Logical Consistency of Natural Language Inference"
_EMNLP/2023/Conference — EMNLP 2023 Findings_

### Official Review · Reviewer_59xA · 2023-08-05

**Soundness:** 2

**Excitement:**

3: Ambivalent: It has merits (e.g., it reports state-of-the-art results, the idea is nice), but there are key weaknesses (e.g., it describes incremental work), and it can significantly benefit from another round of revision. However, I won't object to accepting it if my co-reviewers champion it.

**Paper Topic And Main Contributions:**

The paper suggests a novel method for assessing LLMs beyond their performance on a given task. The authors point out that high performance can stem from recognising spurious correlations and heuristics. The approach proposes to use propositional logic (PL) constructs for assessing the consistency of the models over a variety of logical forms. Such logical variations are achieved by generating negations and combining the original premise hypothesis pairs using an LLM. The authors show that the models are unable to answer ~51% of the rewritten but logically consistent examples.

**Reasons To Accept:**

1) The paper points out a major flaw in the existing LLMs and suggests a novel method for systematically assessing logical discrepancies.
2) A  variety of Proposition Logical constructs are tested for model assessment.
3) The authors suggest that using a set of generated PL examples for tuning the model can lead to higher logical consistency within the model.

**Reasons To Reject:**

The most inherently problematic part of the paper is the negation generation through the use of an LLM (with GPT-3). Although authors suggest that 500 sampled sentences were verified by an author for negation quality, a set of questions still remains.

1) SNLI is a dataset where the premise-hypothesis pair can include several sentences (sentence-pieces). I.E., one of the first premises of SNLI *"This church choir sings to the masses as they sing joyous songs from the book at a church".* A negation of a sentence here is not as straightforward as we would require for PL. A valid negation can cover only one aspect of the premise but still be a direct negation.

i.e.
1) "This church choir sings to the masses as they sing joyous songs from the book at **somewhere other than church.**"
2) *"This church choir does not sing to the masses as they do not sing joyous songs from the book at a church".*

Both of these examples are valid negations of one aspect in the sentence; however, they impact greatly when predicting the label of the premise-hypothesis pair. (i.e. if the hypothesis is "The church is filled with song." )

2) Generating negations is bound to produce random hallucinations within the generation bound, which can, in turn, impact the final prediction, thus hindering the use of strict propositional logic.

3) Dealing with strict logical conjunctions and disjunctions is not as straightforward as adding "and/or" or concatenating two sentences. This part has to be analysed/justified in a deeper manner.

4) Why is only SNLI used for such an evaluation? Is it enough for a definitive conclusion ?


**Reproducibility:**

4: Could mostly reproduce the results, but there may be some variation because of sample variance or minor variations in their interpretation of the protocol or method.

**Reviewer Confidence:**

4: Quite sure. I tried to check the important points carefully. It's unlikely, though conceivable, that I missed something that should affect my ratings.

---

### Official Review · Reviewer_jb1M · 2023-08-05

**Soundness:** 3

**Excitement:**

3: Ambivalent: It has merits (e.g., it reports state-of-the-art results, the idea is nice), but there are key weaknesses (e.g., it describes incremental work), and it can significantly benefit from another round of revision. However, I won't object to accepting it if my co-reviewers champion it.

**Paper Topic And Main Contributions:**

This paper explores nine different data augmentation ways that could be used to fool a well-trained NLI model. The main contribution lies in extensively testing how different forms/combinations of the original premise and hypothesis can fool NLI models. The tested models include RoBERTa, BART, FLAN-T5, and GPT series.

It is more like an NLP engineering experiment paper.

This paper reveals the fact that existing models are not robust to different forms/combinations of the existing premise and hypothesis, which could consider to use as data augmentation in future to train a robust NLP model.

**Reasons To Accept:**

The authors make extensive experiments to see if the new format of the original premise and hypothesis still leads to the same predict.

As I mentioned above, six different backbones are tested under single-task, multi-task, prompting three different scenarios. The experiments are sufficient especially considering this is a short paper.

**Reasons To Reject:**

In attack papers, stealthiness testing is typically essential, as the fundamental hypothesis is that the modified data should closely resemble the original data. This similarity is crucial to prevent people from easily detecting that the training data has been manipulated. However, in this paper, the modified data appears to be significantly different from the original data, which challenges this principle. Moreover, one could argue that deleting every token in the model could also result in a high attack success rate, making it essential to carefully assess and justify the methods used in the paper to maintain a meaningful evaluation of the attack's stealthiness.

**Reproducibility:**

5: Could easily reproduce the results.

**Reviewer Confidence:**

4: Quite sure. I tried to check the important points carefully. It's unlikely, though conceivable, that I missed something that should affect my ratings.

---

### Official Review · Reviewer_n17E · 2023-08-11

**Soundness:** 3

**Excitement:**

3: Ambivalent: It has merits (e.g., it reports state-of-the-art results, the idea is nice), but there are key weaknesses (e.g., it describes incremental work), and it can significantly benefit from another round of revision. However, I won't object to accepting it if my co-reviewers champion it.

**Paper Topic And Main Contributions:**

Authors propose a method to create adversarial attacks on LLMs through perturbed samples, reducing model performance on SNLI (subset with label == entailment). The authors outline 6 rules to transform a premise and hypothesis then attempt to find examples where a NLI model fails to predict the correct entailment. Authors evaluate their methodology on RoBERTa, BART, GPT3/3.5/4 and T5. Discussion by the Authors conclude that large models are less prone to attacks, smaller models are susceptible (76% RoBERTa ASR vs 20% GPT4 ASR). Authors then propose a solution to fine tune models on successful attack samples, claiming a lower ASR (0.7%).

**Questions For The Authors:**

Question A) Why is there a large improvement in mitigating ASR for ChatGPT/GPT-4 over FLAN-T5/GPT3?
Question B) How did you evaluate the quality of perturbations such that they adhere to the correct logic?
Question C) What sampling parameters or temperature settings did you use for LLM models?

**Reasons To Accept:**

The authors propose 9 perturbation strategies from propositional calculus to craft adversarial attack samples to 7 models. The evaluations are provided clearly and the contributions the authors claim are clearly described. The appendix is detailed to provide the necessary context for each strategy. Prompts are clearly enumerated in appendix. Well focused problem statement and approach for a short paper.

**Reasons To Reject:**

With the amount of content provided in the appendix and descriptions related to attack strategies - readers may not understand at first glance the purpose of each adversarial attack. Figure 1 is not legible (no reference numbers to differentiate baseline vs proposed for Accuracy). No discussion is placed on the quality of perturbations vs failures via prompting. There is no clear method (except manual inspection) to ensuring LLM generation adheres to negation logic, simple construction of perturb answers could have influenced results in larger models (Authors provide 0/3/6-shot prompts for entailment tasks & negation generation).

**Reproducibility:**

4: Could mostly reproduce the results, but there may be some variation because of sample variance or minor variations in their interpretation of the protocol or method.

**Reviewer Confidence:**

5: Positive that my evaluation is correct. I read the paper very carefully and I am very familiar with related work.

**Typos Grammar Style And Presentation Improvements:**

Figure 1 can have either numbers reported or highlight the difference between baseline and proposed bars.
Appendix C.1 sentences - the formatting for sentence/negation could be clearer by bolding the labels for the reader.
I found it initially difficult to follow the attack examples and their relationship to the formal expressions - the authors could clarify this through additional whitespace or formatting to make it easier for readers.

---

### Meta-Review · Area_Chair_bQzt · 2023-09-17

**Recommendation:** 3

**Metareview:**

This adversarial-attack paper describes how perturbations can significantly affect accuracy when well-known models like RoBERTa, BART, FLAN-T5, and GPT do natural-language inference tasks, showing that there is a lack of logical consistency in these models.

Pros:
- The problem statement is well-focused, and the approach and scope is appropriate for a short paper.
- There is sufficient information to reproduce this work; the data and source code will be openly available.
- There are extensive and detailed experiments.

Cons:
- The paper uses an LLM to generate the perturbations, which may cause some to doubt their quality; but the authors assessed the quality of a sample of these generated perturbations and found no hallucinations.
- The review copy of the paper was somewhat hard to follow in places due to its heavy reliance on the appendix, which was itself hard to follow in places due to the formatting. Restructuring and/or reformatting somewhat may improve readability.

---

### Decision · Program_Chairs · 2023-10-07

**Decision:**

Accept-Findings

**Comment:**

This adversarial-attack paper describes how perturbations can significantly affect accuracy when well-known models like RoBERTa, BART, FLAN-T5, and GPT do natural-language inference tasks, showing that there is a lack of logical consistency in these models.

Pros:
- The problem statement is well-focused, and the approach and scope is appropriate for a short paper.
- There is sufficient information to reproduce this work; the data and source code will be openly available.
- There are extensive and detailed experiments.

Cons:
- The paper uses an LLM to generate the perturbations, which may cause some to doubt their quality; but the authors assessed the quality of a sample of these generated perturbations and found no hallucinations.
- The review copy of the paper was somewhat hard to follow in places due to its heavy reliance on the appendix, which was itself hard to follow in places due to the formatting. Restructuring and/or reformatting somewhat may improve readability.